# Tau and Aβ$_{42}$ in lavage fluid of pneumonia patients are associated with end-organ dysfunction: A prospective exploratory study

Phoibe Renema[1,2,3☯], Jean-Francois Pittet[4☯], Angela P. Brandon[4], Sixto M. Leal, Jr.[5], Steven Gu[4], Grace Promer[4], Andrew Hackney[4], Phillip Braswell[4], Andrew Pickering[4], Grace Rafield[4], Sarah Voth[6], Ron Balczon[1,7], Mike T. Lin[1,2], K. Adam Morrow[6], Jessica Bell[1,2], Jonathon P. Audia[1,8], Diego Alvarez[9], Troy Stevens[1,2,10‡], Brant M. Wagener[4‡]*

1 Center for Lung Biology, University of South Alabama, Mobile, Alabama, United States of America, 2 Department of Physiology and Cell Biology, University of South Alabama, Mobile, Alabama, United States of America, 3 Department of Biomedical Sciences, University of South Alabama, Mobile, Alabama, United States of America, 4 Department of Anesthesiology and Perioperative Medicine, University of Alabama at Birmingham, Birmingham, Alabama, United States of America, 5 Department of Pathology, University of Alabama at Birmingham, Birmingham, Alabama, United States of America, 6 Department of Cell Biology and Physiology, Edward Via College of Osteopathic Medicine, Monroe, Louisiana, United States of America, 7 Department of Biochemistry and Molecular Biology, University of South Alabama, Mobile, Alabama, United States of America, 8 Department of Microbiology and Immunology, University of South Alabama, Mobile, Alabama, United States of America, 9 Department of Physiology and Pharmacology, Sam Houston State University, Conroe, Texas, United States of America, 10 Department of Internal Medicine, University of South Alabama, Mobile, Alabama, United States of America

☯ These authors contributed equally to this work.
‡ TS and BMW also contributed equally to this work.
* bwagener@uabmc.edu

**Data Availability Statement:** All relevant data are within the manuscript and its Supporting information files.

## Abstract

### Background

Bacterial pneumonia and sepsis are both common causes of end-organ dysfunction, especially in immunocompromised and critically ill patients. Pre-clinical data demonstrate that bacterial pneumonia and sepsis elicit the production of cytotoxic tau and amyloids from pulmonary endothelial cells, which cause lung and brain injury in naïve animal subjects, independent of the primary infection. The contribution of infection-elicited cytotoxic tau and amyloids to end-organ dysfunction has not been examined in the clinical setting. We hypothesized that cytotoxic tau and amyloids are present in the bronchoalveolar lavage fluid of critically ill patients with bacterial pneumonia and that these tau/amyloids are associated with end-organ dysfunction.

### Methods

Bacterial culture-positive and culture-negative mechanically ventilated patients were recruited into a prospective, exploratory observational study. Levels of tau and Aβ$_{42}$ in, and cytotoxicity of, the bronchoalveolar lavage fluid were measured. Cytotoxic tau and amyloid concentrations were examined in comparison with patient clinical characteristics, including measures of end-organ dysfunction.

**Funding:** This study was supported by: B.M.W.–GM127584 and GM127584-S1; National Institute of General Medical Sciences; https://www.nigms.nih.gov T. S. and R.B.–HL66299 and HL148069; National Heart, Lung, and Blood Institute; https://www.nhlbi.nih.gov M.L., T.S., R.B.–HL140182; National Heart, Lung, and Blood Institute; https://www.nhlbi.nih.gov J.P.A. and D.A.–HL118334; National Heart, Lung, and Blood Institute; https://www.nhlbi.nih.gov S.M.L–AI170719; National Institute of Allergy and Infectious Diseases; https://www.niaid.nih.gov S.V.–HL147512, HL007778, REAP220049A0001; National Heart, Lung, and Blood Institute; https://www.nhlbi.nih.gov and Edward Via College of Osteopathic Medicine Research Eureka Accelerator Program (REAP); https://www.vcom.edu/research/research-strategic-plan None of the sponsors or funders played any role in the study design, data collection and analysis, decision to publish, or preparation of the manuscript. There was no additional external funding received for this study.

**Competing interests:** The authors have declared that no competing interests exist.

## Results

Tau and $A\beta_{42}$ were increased in culture-positive patients (n = 49) compared to culture-negative patients (n = 50), independent of the causative bacterial organism. The mean age of patients was 52.1 ± 16.72 years old in the culture-positive group and 52.78 ± 18.18 years old in the culture-negative group. Males comprised 65.3% of the culture-positive group and 56% of the culture-negative group. Caucasian culture-positive patients had increased tau, boiled tau, and $A\beta_{42}$ compared to both Caucasian and minority culture-negative patients. The increase in cytotoxins was most evident in males of all ages, and their presence was associated with end-organ dysfunction.

## Conclusions

Bacterial infection promotes the generation of cytotoxic tau and $A\beta_{42}$ within the lung, and these cytotoxins contribute to end-organ dysfunction among critically ill patients. This work illuminates an unappreciated mechanism of injury in critical illness.

## Introduction

Bacterial pneumonia and sepsis are common causes of critical illness. The host response to infection plays a cardinal role in injury propagation, oftentimes resulting in organ dysfunction peripheral to the lung, including the heart, brain, and kidney [1–4]. Yet, at present, efforts to thwart a dysregulated immune response have failed to reduce the incidence or severity of end-organ dysfunction, and have not uniformly improved patient outcomes [5–12]. Hence, a better understanding of the host response to infection is necessary to develop novel medical therapies that improve outcomes of critically ill patients.

Recently, we have discovered that pneumonia elicits lung endothelial production of cytotoxic tau and amyloids [13, 14] and reviewed in [15]. These lung-derived cytotoxins are similar to those found in the brain [16, 17]. Indeed, tau and amyloids are well-recognized for their contribution to many neurocognitive disorders, such as Alzheimer's disease, frontotemporal tauopathies, chronic traumatic encephalopathy, and Parkinson's disease [18–21]. Furthermore, amyloids have been implicated as a potential mechanism of injury in diabetes and other amyloidopathies [22–24]. Interestingly, Alois Alzheimer suggested an "infection hypothesis" for an initial source of cytotoxic amyloids, but this hypothesis has only recently regained traction among scientists [25–27].

Pre-clinical data demonstrate that cytotoxic tau and amyloids are elicited from pulmonary endothelial cells after infection with *Pseudomonas aeruginosa* in a Type III secretion system-dependent manner [13, 17]. The cytotoxic tau and amyloids injure naïve endothelial cells, and cause lung, heart, and brain injury independently of the primary infection [13, 14, 16]. Furthermore, bronchoalveolar lavage fluid and cerebrospinal fluid from patients with bacterial pneumonia contribute to tau- and amyloid-dependent lung [13] and brain injury in mice [14], respectively, whereas uninfected patient samples do not cause end-organ dysfunction. Finally, HBSS washes from extra-corporeal membrane oxygenation oxygenators of infected patients, but not uninfected patients, contain tau and amyloids that are toxic to naïve animals [16]. While these results illustrate the contribution of infection-elicited cytotoxic tau and amyloids to end-organ dysfunction in pre-clinical animal models [15], the infectious proteinopathy hypothesis has not been tested in a prospective clinical study.

To determine whether cytotoxic tau and amyloids are present in the bronchoalveolar lavage fluid of critically ill patients with bacterial pneumonia, and whether these cytotoxins contribute to end-organ dysfunction, we conducted a prospective, exploratory observational study. We recruited mechanically ventilated patients who were either culture positive or culture negative for bacterial pneumonia. Bronchoalveolar lavage fluid was collected and tau, $A\beta_{42}$, and their relative cytotoxicity were measured. Clinical characteristics and outcomes were evaluated for these patients. Our results indicate that bacterial pneumonia promotes the generation of lung-derived cytotoxic tau and amyloids that are associated with end-organ dysfunction.

## Methods

### Study design and determination of end-organ dysfunction

We conducted a single-center, exploratory, prospective, controlled observational study at the University of Alabama at Birmingham Hospital between June 1, 2016 and March 15, 2019. Patients were recruited from four intensive care units: Trauma-Burn, Neurological, Surgical, or Cardiothoracic Intensive Care Units (ICU). All mechanically ventilated patients with a bronchoalveolar lavage (BAL) fluid containing >100,000 colony-forming units (CFU) of bacteria or "no growth" were considered. Exclusion criteria included age <18 years, any *prior* positive culture from any source (in either the positive or negative groups), incarceration, pregnancy, patients enrolled in an ongoing, interventional clinical trial, or consent refusal. This study was conducted in accordance with the Declaration of Helsinki and the protocol was approved by The University of Alabama at Birmingham Institutional Review Board (Protocol Number—150209001). Written or verbal (via phone) consent was obtained from the patient or the patient's Legally Authorized Representative. If verbal consent was obtained, we followed up with each patient or patient's Legally Authorized Representative to obtain formal written consent. If this secondary written consent could not be obtained, patients were excluded from the study. Patients were followed from ICU admission to ICU discharge and clinical data were collected. Patients were examined for acute kidney injury, cardiovascular dysfunction, and coagulopathy after recruitment for two weeks. For acute kidney injury, we used the Kidney Disease Improving Global Outcomes definition for creatinine only [28]. A diagnosis of acute kidney injury was a serum creatinine increased by 0.3mg/dL over any 48-hour period or the patient was started on renal replacement therapy. For cardiovascular dysfunction, we evaluated all patients for the presence of a vasopressor or inotrope after recruitment. A diagnosis of coagulopathy was an international normalized ratio (INR) of $\geq$1.2 after recruitment. Some patients did not have their INR measured during the study and were excluded from this particular analysis.

### Sample preparation

Bronchoalveolar lavage (BAL) fluid was obtained as previously described [13]. Briefly, BAL samples were centrifuged (5000 x *g*, 10', 4˚C), filter sterilized (0.22μμm), and stored at -80˚C until assessment.

### Measurement of amyloid

Amyloid concentrations in BAL fluid samples were measured using a commercially available Amyloid beta 42 Human ELISA Kit (ThermoFisher, #KHB3441) according to manufacturer instructions. Each BAL fluid sample was assessed in duplicate. The assay range is 15.6–1000 pg/mL and samples with a value below 15.6 pg/mL are "non-responders". All others are "responders".

## Measurement of tau

Tau concentrations in BAL fluid samples were measured using an in-house ELISA with T22 antibody as previously described [16]. Cytotoxic tau has previously been shown to be heat stable [13]. Therefore, BAL fluid samples were divided into two aliquots and either boiled (100˚C, 20') or not. PA103 infection-elicited tau production from rat pulmonary microvascular endothelial cells was used as a positive control. T22 immunoreactivity values obtained from BAL fluid were represented as a fold-change over T22 immunoreactivity values obtained from PA103-infected rat pulmonary microvascular endothelial cell supernatants. "Non-responders" had a fold-change <1 and "responders" had a fold-change ≥1.

## Measurement of bronchoalveolar lavage fluid cytotoxicity

To determine the cytotoxicity of tau/amyloid in the BAL fluid, rat pulmonary microvascular endothelial cell lysis following exposure to BAL fluid was measured using the CyQUANT™ LDH cytotoxicity assay (ThermoFisher #C20300). Confluent naïve rat pulmonary microvascular endothelial cells in a 96-well culture dish were incubated with 50μμL BAL fluid and 50μμL HBSS (ThermoFisher #14025092) for 18 hours. Supernatants were assessed for lactate dehydrogenase release in triplicate according to manufacturer instructions.

## Statistical analysis

All demographic and clinical variables with continuous measures were expressed as mean ± SD and categorical variables were expressed as proportions. Tau, boiled tau, $A\beta_{42}$, and cytotoxicity are expressed as mean ± SEM. For normally distributed data, we used a Student's t-test to compare two groups or a one-way ANOVA with post-hoc Tukey's multiple comparisons test to compare three or more groups. Statistical analyses were done with Prism Graph-Pad 9.0. A $p$ value of ≤0.05 was considered statistically significant.

# Results

## Patient recruitment and demographics

Bacterial pneumonia is a common cause of end-organ dysfunction. Our prior pre-clinical studies have suggested that tau and $A\beta_{42}$ may contribute to acute and long-term morbidity and mortality in critically ill patients [13–17]. Therefore, we designed a prospective, exploratory observational study to determine whether bacterial pneumonia elicits release of tau and $A\beta_{42}$ in the bronchoalveolar lavage fluid and whether the presence of tau and $A\beta_{42}$ portends end-organ dysfunction. Bronchoalveolar lavage fluid was collected from mechanically ventilated patients. Patients were recruited to the "culture-negative" (CN) group if there was no growth in their BAL fluid cultures, and were recruited to the "culture-positive" (CP) group if there were >100,000 CFU of one or more bacterial species, as detailed in the *Methods*. Of 171 patients screened, 99 patients were recruited in total, including 50 to the CN group and 49 to the CP group (Fig 1). Table 1 shows the demographic characteristics of recruited patients. The mean age was 52 years and 50 patients were male. Sixty-five percent were Caucasian, 32% were African-American, and 3% were Hispanic. Forty-one percent of patients consumed alcohol regularly, 46% were smokers, and 12% had a history of illicit drug use. There were no demographic differences between CN and CP patients, except for illicit drug use, which was significantly increased in the CP group. At the time of study recruitment, the median Acute Physiology and Chronic Health Evaluation II score, 4-point Lung Injury Score, Sepsis-related Organ Failure Assessment, and Simplified Acute Physiology Score II scores in the total cohort were 13, 1, 5, and 33, respectively (Table 2). Patients were in the hospital for 24 days, in the

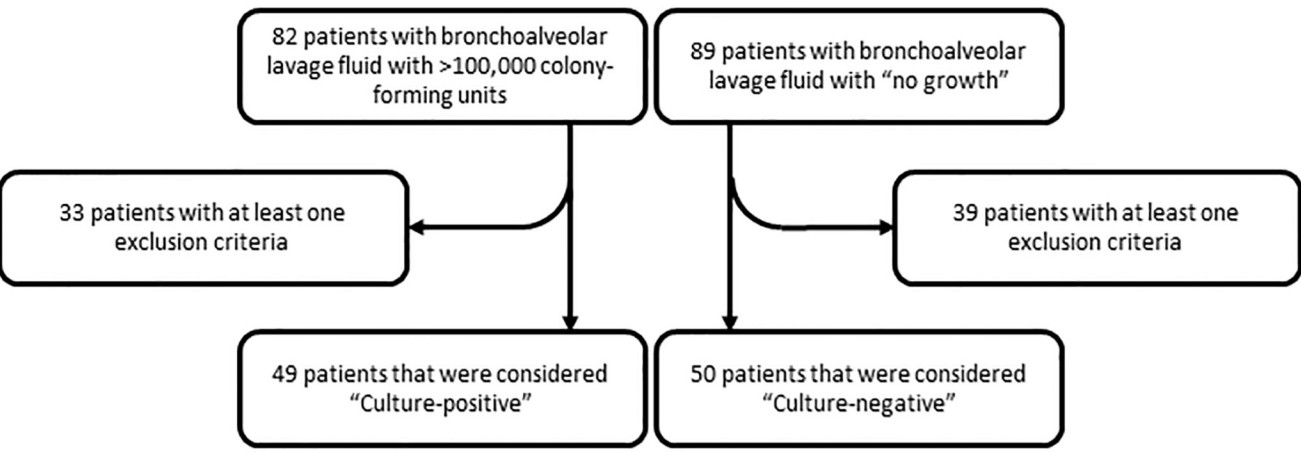

**Fig 1. Flowchart of patient screening.** The number of patients screened, excluded, and recruited to the study are listed for both the culture-positive and culture-negative arms.

ICU for 18 days, and mechanically ventilated for 13 days in the entire cohort. Clinical characteristics were not different between CN and CP groups with the exception of the lung injury score, which was significantly higher in the CP group.

## Tau and A$\beta_{42}$ levels are increased in the bronchoalveolar lavage fluid of culture-positive patients

Tau and A$\beta_{42}$ levels were measured in the bronchoalveolar lavage fluid of CN and CP patients (Fig 2). Tau (Fig 2A), boiled tau (Fig 2B), and A$\beta_{42}$ (Fig 2C) were significantly increased in the CP group compared to the CN group. Of note, there was not enough bronchoalveolar lavage fluid for every measurement in all patients reflected in the "n" below each group. We have previously shown that infection-induced tau and amyloids are cytotoxic both *in vitro* and *in vivo* [13, 16]. To determine whether bronchoalveolar lavage fluid samples exhibited cytotoxicity, potentially due to amyloids, we conducted an *in vitro* cytotoxicity assay. Cytotoxicity of the CP group was significantly higher than the CN group (Fig 2D), consistent with the presence of cytotoxic tau and A$\beta_{42}$.

We next questioned whether the increase in cytotoxic tau, boiled tau, and A$\beta_{42}$ in CP bronchoalveolar lavage fluid was bacterial species-dependent. We compared tau, boiled tau, A$\beta_{42}$, and cytotoxicity to the species of bacteria (S1 Fig) and Gram stain classification (S2 Fig). When divided among the three most common species found in CP patients in our study (*Pseudomonas aeruginosa*, *Staphylococcus aureus*, *Klebsiella pneumoniae*), other single bacterial cultures, and polymicrobial cultures, there were small differences in tau and cytotoxicity levels compared to cultures with multiple microbes. Furthermore, when the levels were compared between Gram-positive bacteria, Gram-negative bacteria, or both, there were no differences. These data indicate that the presence of bacteria in the lung increased cytotoxic tau, boiled tau, and A$\beta_{42}$ in the bronchoalveolar lavage fluid, independent of the bacterial species or Gram stain classification.

## Tau and A$\beta_{42}$ are elevated in the bronchoalveolar lavage fluid of males

Next, we examined tau, boiled tau, A$\beta_{42}$, and lavage fluid cytotoxicity, divided by sex (Fig 3A–3D), age (Fig 3E–3H), and race (Fig 3I–3L). Male CP patients had increased tau, boiled tau,

**Table 1. Patient demographic characteristics.**

| Characteristics | Total | Culture-Positive | Culture-Negative | *p*-value |
|---|---|---|---|---|
| **Total Patients** | 99 | 49 | 50 | n/a |
| **Age in Years (Avg ± SD)** | 52.44 ± 17.39 | 52.1 ± 16.72 | 52.78 ± 18.18 | 0.8469 |
| **Gender (Male, %)** | 60 (61) | 32 (65) | 28 (56) | 0.4124 |
| **Race** | | | | |
| **Caucasian (n, %)** | 64 (65) | 35 (71) | 29 (58) | 0.2081 |
| **African-American (n, %)** | 32 (32) | 13 (27) | 19 (38) | 0.2837 |
| **Hispanic (n,%)** | 3 (3) | 1 (2) | 2 (4) | 0.9999 |
| **Alcohol Consumer** | | | | |
| **Current (n, %)** | 41 (41) | 18 (37) | 23 (46) | 0.4162 |
| **None (n %)** | 49 (50) | 27 (55) | 22 (44) | 0.3173 |
| **Former (n, %)** | 8 (8) | 4 (8) | 4 (8) | 0.9999 |
| **Unknown (n, %)** | 1 (1) | 0 (0) | 1 (2) | 0.9999 |
| **Smoking** | | | | |
| **Current (n, %)** | 45 (46) | 26 (53) | 19 (38) | 0.1598 |
| **None (n %)** | 40 (40) | 21 (43) | 19 (38) | 0.6845 |
| **Former (n, %)** | 13 (13) | 2 (4) | 11 (22) | 0.0147 |
| **Unknown (n, %)** | 1 (1) | 0 | 1 (2) | 0.9999 |
| **Illicit Drug Use** | | | | |
| **Current (n, %)** | 12 (12) | 12 (25) | 0 (0) | <0.0001 |
| **None (n %)** | 72 (73) | 27 (55) | 45 (90) | 0.0001 |
| **Former (n, %)** | 13 (13) | 9 (18) | 4 (8) | 0.1478 |
| **Unknown (n, %)** | 2 (2) | 1 (2) | 1 (2) | 0.9999 |

and $A\beta_{42}$ compared to both male and female CN patients (Fig 3A–3C), and female CP patients exhibited increased cytotoxicity compared to both male and female CN patients (Fig 3D). Overall, male patients in the cohort exhibited the most prominent increase in bronchoalveolar lavage fluid concentrations of cytotoxic tau and amyloids (Fig 3A–3C). For age analysis, we defined "younger" as <50 years and "older" as ≥50 years. Older CP groups had increased tau compared to older CN groups (Fig 3E), while younger CP patients had increased tau, boiled tau, and $A\beta_{42}$ compared to older CN patients (Fig 3E–3G) and increased $A\beta_{42}$ compared to younger CN patients (Fig 3G). Finally, due to low numbers of Hispanic patients in our study, race was categorized into Caucasian and minority groups for analysis. Caucasian CP patients

**Table 2. Patient clinical characteristics.**

| Injury Scores | Median (Range) | Culture-Positive (Avg ± SD) | Culture-Negative (Avg ± SD) | *p*-value |
|---|---|---|---|---|
| **APACHE II** | 13 (0–38) | 14.55 ± 7.25 | 13.58 ± 6.67 | 0.49 |
| **4 Point Lung Injury** | 1 (0–2.7) | 1.20 ± 0.59 | 0.79 ± 0.68 | 0.0018 |
| **SOFA** | 5 (1–18) | 5.27 ± 2.88 | 5.10 ± 2.43 | 0.7514 |
| **SAPS II** | 33 (13–74) | 36.63 ± 12.75 | 32.52 ± 9.59 | 0.0726 |
| **Characteristics** | | | | |
| **Length of Hospital Stay (Days)** | 24 (2–108) | 27.80 ± 19.71 | 30.68 ± 20.63 | 0.4794 |
| **Length of ICU Stay (Days)** | 18 (2–106) | 20.41 ± 16.99 | 20.34 ± 15.39 | 0.9829 |
| **Duration of Mechanical Ventilation (Days)** | 13 (1–61) | 14.78 ± 12.14 | 15.36 ± 13.70 | 0.8242 |

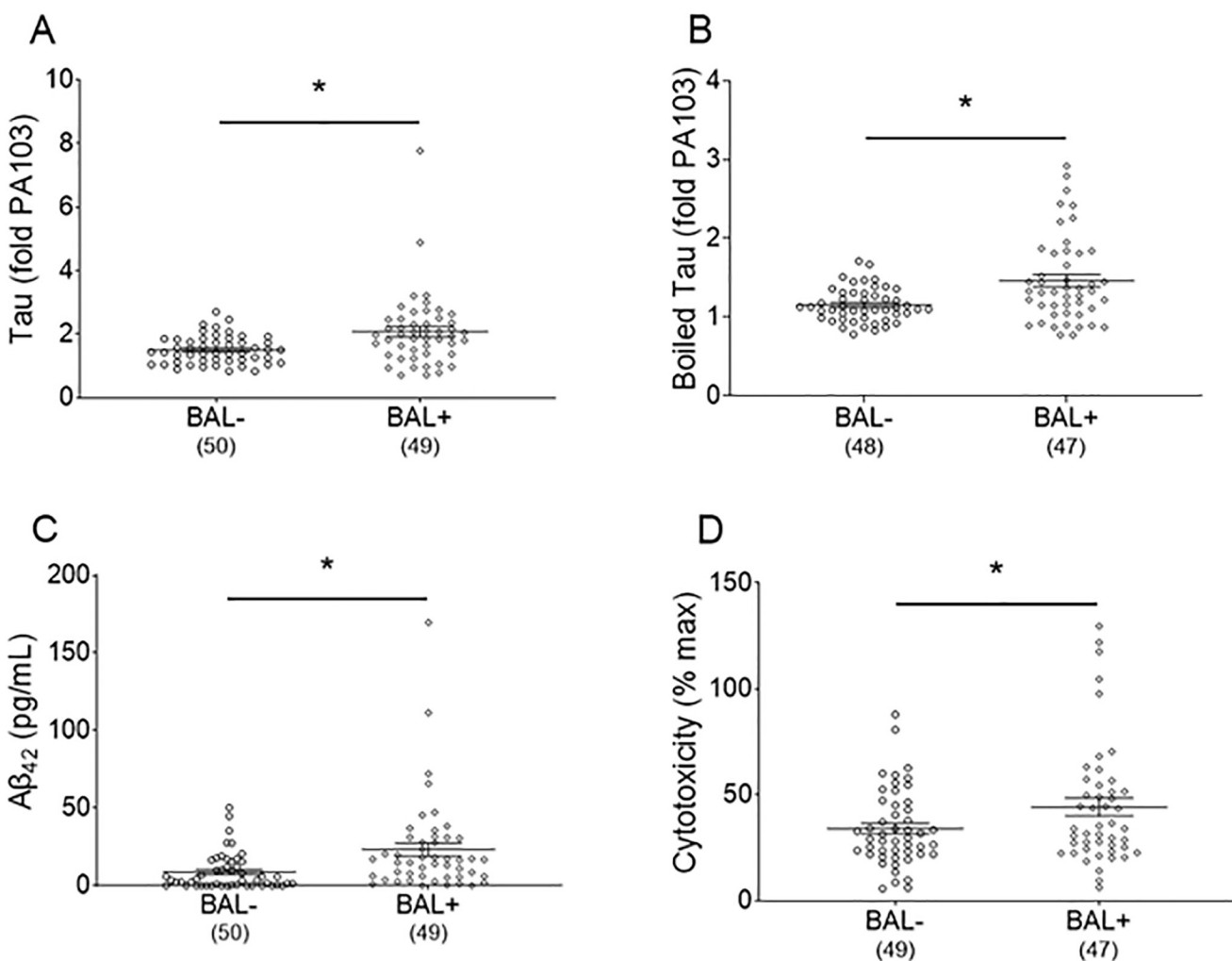

**Fig 2. Tau, boiled tau, Aβ₄₂, and cytotoxicity levels in culture-positive and culture-negative patient lavage fluid.** *(A)* Tau, *(B)* boiled tau, *(C)* Aβ₄₂, and *(D)* cytotoxicity were assayed as described in the *Methods* for culture-positive and culture-negative patients. Data are expressed as mean ± SEM, *n* are listed below groups; * represents $p < 0.05$.

had increased tau, boiled tau, and Aβ₄₂ compared to both Caucasian and minority CN groups (Fig 3I–3K).

## Tau and Aβ₄₂ in the bronchoalveolar lavage fluid are associated with end-organ dysfunction but not mortality

We examined whether tau, boiled tau, Aβ₄₂, and cytotoxicity were associated with indices of end-organ dysfunction and mortality. We analyzed CN and CP patients for evidence of acute kidney injury (Fig 4A–4D), coagulopathy (Fig 4E–4H), and vasopressor use (Fig 4I–4L). First, tau and boiled tau levels were higher in CP patients without acute kidney injury compared to both CN patients with and without acute kidney injury (Fig 4A and 4B). Aβ₄₂ was increased in CP patients with acute kidney injury compared to both CN patients with and without acute kidney injury (Fig 4C). For coagulopathy, tau, boiled tau, and Aβ₄₂ were all increased in CP patients with coagulopathy compared to CN patients with coagulopathy (Fig 4E–4G).

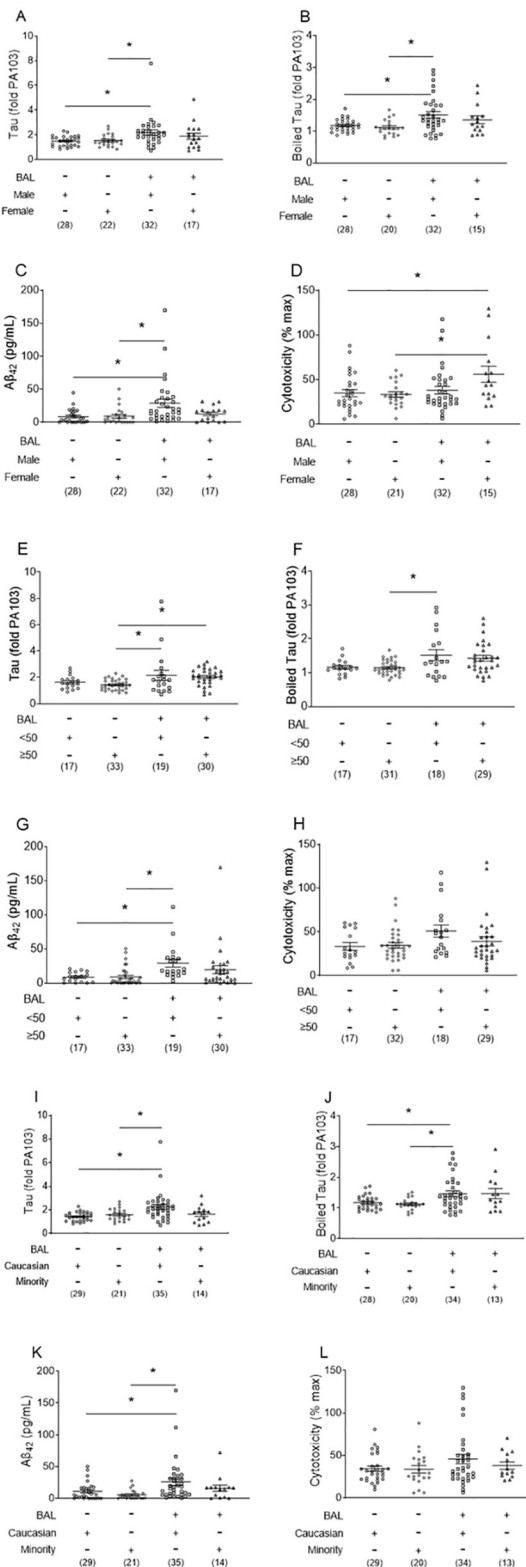

**Fig 3. Tau, boiled tau, Aβ₄₂, and cytotoxicity levels in lavage fluid are different in sex, age, and race of culture-positive and culture-negative patients.** *(A, E, I)* Tau, *(B, F, J)* boiled tau, *(C, G, K)* Aβ₄₂, and *(D, H, L)* cytotoxicity were assayed as described in the Methods for *(A-D)* sex, *(E-H)* age, and *(I-L)* race of culture-positive and culture-negative patients. Data are expressed as mean ± SEM, *n* are listed below groups; * represents $p < 0.05$.

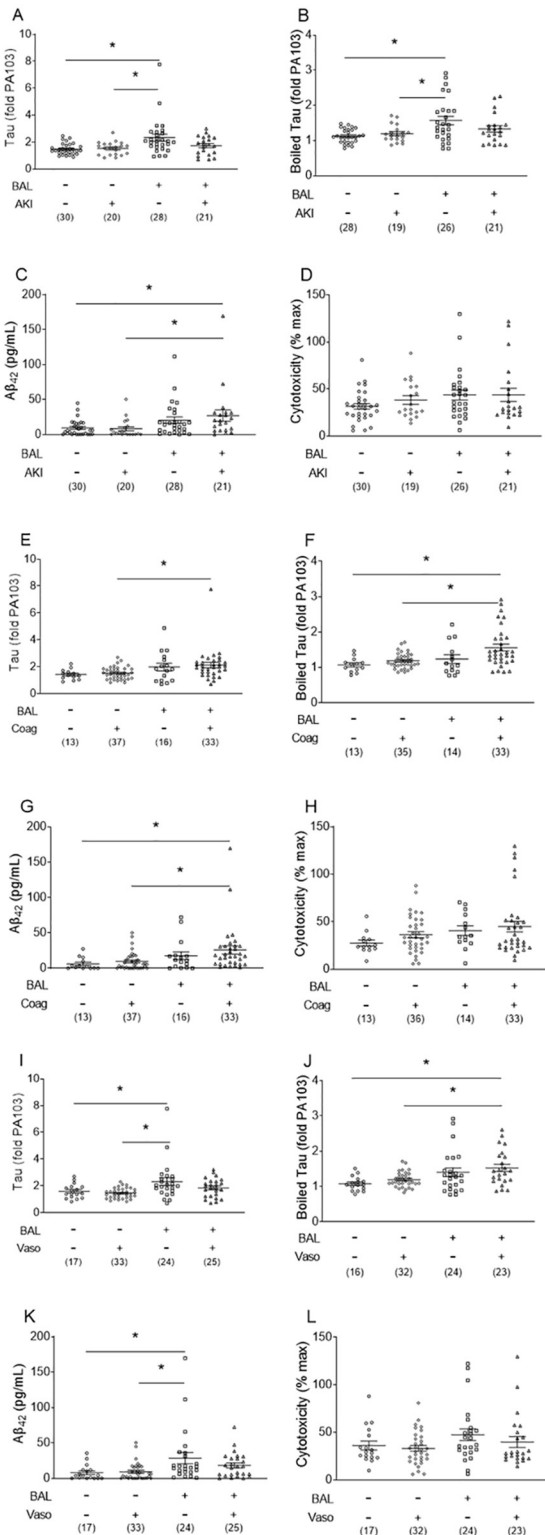

**Fig 4. Tau, boiled tau, Aβ₄₂, and cytotoxicity levels are different in end organ dysfunction groups of culture-positive and culture-negative patients.** *(A, E, I)* Tau, *(B, F, J)* boiled tau, *(C, G, K)* Aβ₄₂, and *(D, H, L)* cytotoxicity were assayed as described in the Methods for *(A-D)* acute kidney injury, *(E-H)* coagulopathy, and *(I-L)* cardiovascular dysfunction of culture-positive and culture-negative patients. Data are expressed as mean ± SEM, *n* are listed below groups; * represents $p < 0.05$.

Furthermore, boiled tau and $A\beta_{42}$ were increased in CP patient with coagulopathy compared to CN patients without coagulopathy (Fig 4F and 4G). For vasopressor/inotropes, tau and $A\beta_{42}$ were increased in CP patients without vasopressors compared to both CN patients with and without vasopressors (Fig 4I and 4K). Boiled tau was increased in CP patients with vasopressors compared to CN patients with and without vasopressors (Fig 4J). Thus, bronchoalveolar lavage fluid levels of cytotoxic tau and amyloid were increased in CP patients with acute kidney injury, dysregulated coagulation, and hemodynamic instability.

As an extension of end-organ dysfunction, we examined mortality (Fig 5A–5D). When evaluating death as an outcome, tau, boiled tau, and $A\beta_{42}$ were increased in the CP group that survived compared to the CN group that survived (Fig 5A–5C). Furthermore, $A\beta_{42}$ was increased in CP patients who survived compared to CP patients that died (Fig 5C). In summary, infection increased tau and $A\beta_{42}$ within the bronchoalveolar lavage fluid and an increase in $A\beta_{42}$ was associated with survival.

### Tau and $A\beta_{42}$ in the bronchoalveolar lavage fluid are associated with lung injury

Because not all CP patients exhibited an increase in tau, boiled tau, $A\beta_{42}$, and cytotoxicity, we examined differences in clinical scores in non-responders and responders from CN or CP groups (Table 3). In the tau, boiled tau, and $A\beta_{42}$ responder groups, patients that were CP had an increased lung injury score when compared to CN patients (Table 3B). Thus, when increased tau, boiled tau, and $A\beta_{42}$ levels were found in the lungs of CP patients, there was a higher lung injury score, indicating that tau and $A\beta_{42}$ are not only elicited by bacterial infection, but are associated with increased lung damage.

## Discussion

Bacterial pneumonia is a principal cause of the acute respiratory distress syndrome (ARDS) and sepsis. Critically ill patients often suffer end-organ dysfunction, but the infection-dependent mechanisms triggered within the lung that cause acute and chronic injury of peripheral organs remain poorly understood. Aside from decreased tidal volumes and improved critical care management, adjunct medical therapy has not improved outcomes in ARDS patients, illustrating a lapse in our understanding of the mechanisms responsible for disease progression [6, 29, 30]. We have recently identified a novel host response to pneumonia that may contribute to end-organ dysfunction and offers a novel therapeutic target. Pneumonia elicits production of cytotoxic tau and amyloids within the lung; these cytotoxins are heat and protease stable, are transmissible between cells and animals, consistent with prion disease, and are sufficient to injure naïve organs, including the lung, heart, and brain [13–17]. While these prior studies demonstrated that pneumonia initiates a proteinopathy of the lung, this principle has not been tested in a controlled patient cohort. Here, we hypothesized that critically ill pneumonia patients possess increased levels of cytotoxic tau and $A\beta_{42}$ within the bronchoalveolar lavage fluid and that these airway cytotoxins associate with clinical measures of end-organ dysfunction. We designed a prospective, exploratory observational clinical study in which mechanically ventilated patients were recruited into the study and categorized as either CN or CP, based upon microbiological evidence of bacteria within the bronchoalveolar lavage fluid. Importantly, for CP patients, this was their first infection in the hospital. Our study revealed that tau, boiled tau, and $A\beta_{42}$ in the bronchoalveolar lavage fluid, and cytotoxicity of the bronchoalveolar lavage fluid, were increased in CP patients compared to CN patients. The increase in these cytotoxins in CP patients was: 1) most prominent in males; 2) positively associated with clinical indices of end-organ dysfunction; 3) not associated with mortality; and, 4)

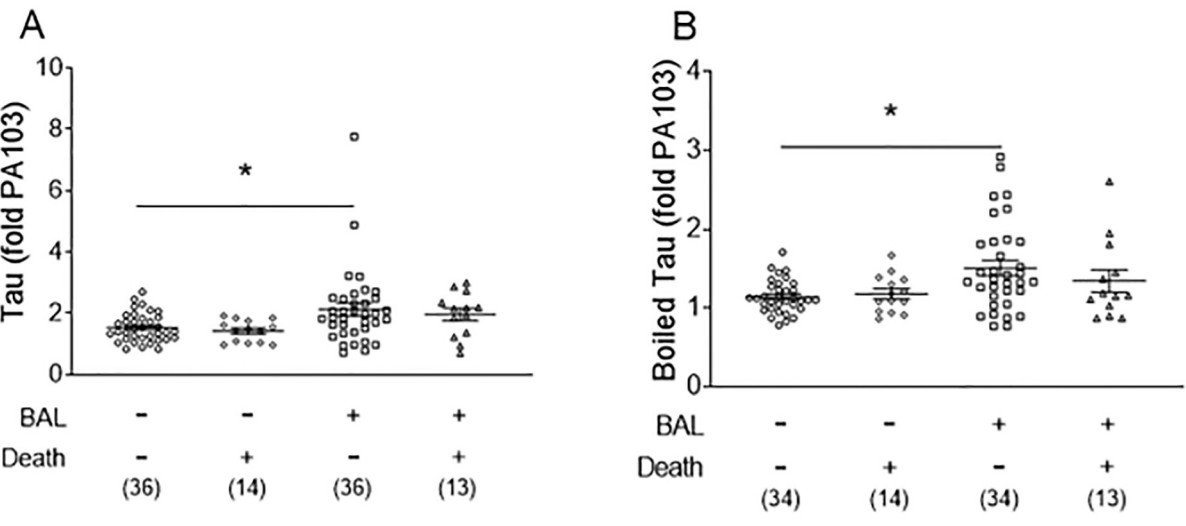

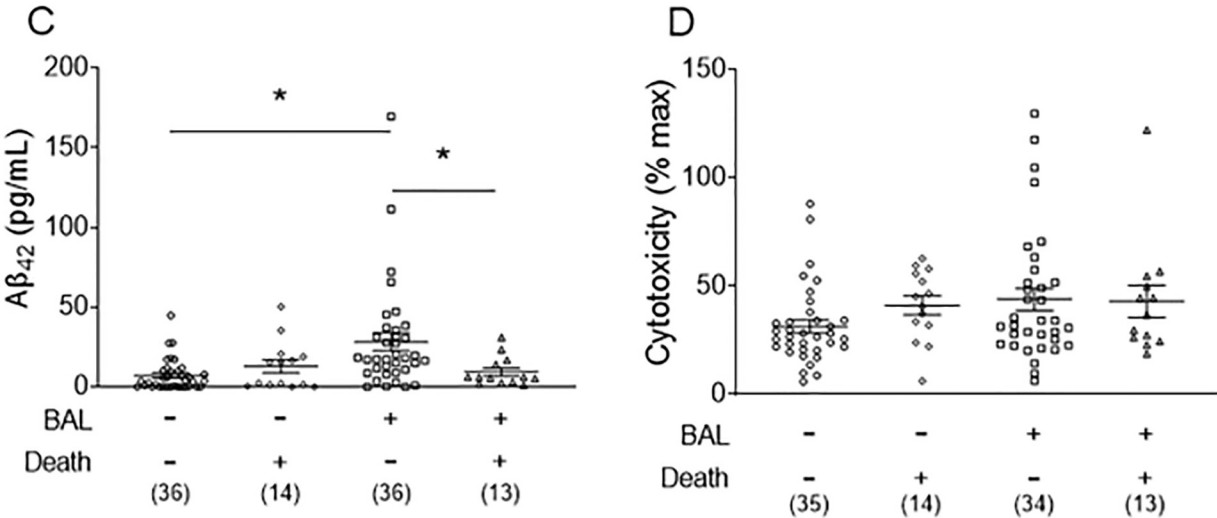

**Fig 5. Tau, boiled tau, Aβ$_{42}$, and cytotoxicity levels are different in mortality of culture-positive and culture-negative patients.** *(A)* Tau, *(B)* boiled tau, *(C)* Aβ$_{42}$, and *(D)* cytotoxicity were assayed as described in the Methods for mortality of culture-positive and culture-negative patients. Data are expressed as mean ± SEM, *n* are listed below groups; * represents $p < 0.05$.

dependent upon bacterial infection, but independent of bacterial species or Gram-stain classification. To our knowledge, this exploratory study is among the first to demonstrate that infection promotes an increase in cytotoxic tau and Aβ$_{42}$ within the lung and that cytotoxin increase may be a determinant of end-organ dysfunction.

**Table 3. Clinical outcomes in biomarker non-responsive and responsive vs culture-negative and culture-positive patients.**

**A. APACHE II**

| Cytotoxin | Non-Responsive, Culture-Negative | Non-Responsive, Culture-Positive | Responsive, Culture-Negative | Responsive, Culture-Positive |
|---|---|---|---|---|
| Tau | 15.83 ± 9.81 (6) | 10.43 ± 5.74 (7) | 13.27 ± 6.22 (44) | 15.24 ± 7.31 (42) |
| Boiled Tau | 15.69 ± 6.66 (13) | 11.56 ± 4.64 (9) | 13.06 ± 6.65 (35) | 15.37 ± 7.71 (38) |
| $A\beta_{42}$ | 13.79 ± 6.64 (39) | 14.67 ± 6.54 (24) | 12.82 ± 7.04 (11) | 14.44 ± 8.02 (25) |

**B. LIS**

| Cytotoxin | Non-Responsive, Culture-Negative | Non-Responsive, Culture-Positive | Responsive, Culture-Negative | Responsive, Culture-Positive |
|---|---|---|---|---|
| Tau | 1.42 ± 0.69 (6)* | 1.11 ± 0.70 (7) | 0.70 ± 0.64 (44)*, ** | 1.21 ± 0.58 (42)** |
| Boiled Tau | 0.95 ± 0.75 (13) | 1.14 ± 0.63 (9) | 0.75 ± 0.66 (35)* | 1.22 ± 0.59 (38)* |
| $A\beta_{42}$ | 0.85 ± 0.71 (39) | 1.15 ± 0.51 (24) | 0.56 ± 0.50 (11)* | 1.25 ± 0.67 (25)* |

**C. SOFA**

| Cytotoxin | Non-Responsive, Culture-Negative | Non-Responsive, Culture-Positive | Responsive, Culture-Negative | Responsive, Culture-Positive |
|---|---|---|---|---|
| Tau | 6.00 ± 2.00 (6) | 4.00 ± 1.29 (7) | 4.98 ± 2.48 (44) | 5.48 ± 3.02 (42) |
| Boiled Tau | 4.62 ± 2.29 (13) | 4.56 ± 2.35 (9) | 5.31 ± 2.53 (35) | 5.37 ± 3.04 (38) |
| $A\beta_{42}$ | 5.03 ± 2.57 (39) | 5.58 ± 3.46 (24) | 5.36 ± 1.96 (11) | 4.96 ± 2.21 (25) |

**D. SAPS II**

| Cytotoxin | Non-Responsive, Culture-Negative | Non-Responsive, Culture-Positive | Responsive, Culture-Negative | Responsive, Culture-Positive |
|---|---|---|---|---|
| Tau | 33.00 ± 11.45 (6) | 30.57 ± 7.02 (7) | 32.45 ± 9.46 (44) | 37.64 ± 13.26 (42) |
| Boiled Tau | 35.46 ± 9.42 (13) | 31.44 ± 10.20 (9) | 31.74 ± 9.67 (35) | 37.79 ± 13.14 (38) |
| $A\beta_{42}$ | 32.46 ± 9.18 (39) | 39.21 ± 13.46 (24) | 32.73 ±11.42 (11) | 34.16 ± 11.77 (25) |

**E. Length of Stay Hospital**

| Cytotoxin | Non-Responsive, Culture-Negative | Non-Responsive, Culture-Positive | Responsive, Culture-Negative | Responsive, Culture-Positive |
|---|---|---|---|---|
| Tau | 33.50 ± 18.02 (6) | 21.29 ± 10.42 (7) | 30.30 ± 21.12 (44) | 28.88 ± 20.75 (42) |
| Boiled Tau | 31.15 ± 18.81; (13) | 19.78 ± 9.62 (9) | 29.37 ± 21.06 (35) | 30.00 ± 21.48 (38) |
| $A\beta_{42}$ | 33.67 ± 21.91 (39) | 28.63 ± 22.55 (24) | 20.09 ± 10.15 (11) | 27.00 ± 16.98 (25) |

**F. Length of Stay ICU**

| Cytotoxin | Non-Responsive, Culture-Negative | Non-Responsive, Culture-Positive | Responsive, Culture-Negative | Responsive, Culture-Positive |
|---|---|---|---|---|
| Tau | 21.33 ± 10.60 (6) | 18.71 ± 12.02 (7) | 20.20 ± 16.02 (44) | 20.69 ± 17.79 (42) |
| Boiled Tau | 18.92 ± 9.60 (13) | 15.11 ± 6.47 (9) | 20.80 ± 17.54 (35) | 21.79 ± 18.83 (38) |
| $A\beta_{42}$ | 21.79 ± 16.59 (39) | 21.33 ± 20.81 (24) | 15.18 ± 8.82 (11) | 19.52 ± 12.68 (25) |

**G. Duration on Mechanical Ventilation**

| Cytotoxin | Non-Responsive, Culture-Negative | Non-Responsive, Culture-Positive | Responsive, Culture-Negative | Responsive, Culture-Positive |
|---|---|---|---|---|
| Tau | 23.00 ± 22.27 (6) | 15.43 ± 11.28 (7) | 14.32 ± 12.13 (44) | 14.67 ± 12.41 (42) |
| Boiled Tau | 16.15 ± 15.69 (13) | 13.00 ± 7.37 (9) | 14.40 ± 12.19 (35) | 15.08 ± 13.26 (38) |
| $A\beta_{42}$ | 16.23 ± 14.67 (39) | 13.00 ± 11.17 (24) | 12.27 ± 9.42 (11) | 16.48 ± 13.00 (25) |

Patient clinical scores are provided for tau/boiled tau/$A\beta_{42}$ "non-responders" or "responders" compared to culture-negative vs culture-positive groups. Non-responders and responders are defined in the Methods. Mean ± SD and *n* are provided. Asterisks (single or multiple) represent groups that are statistically different within a specific clinical outcome.

The first goal of this study was to assess whether tau, boiled tau, $A\beta_{42}$, and cytotoxicity were increased in CP patients. We found that cytotoxic tau, boiled tau, and $A\beta_{42}$ were increased in infected patients, when compared to uninfected, patients. All patients recruited to the study were mechanically ventilated and had similar APACHE II, SOFA, and SAPS II scores (Table 2). Thus, the increase in cytotoxic tau and amyloids was due to bacterial pneumonia, rather than critical illness *per se*.

Both susceptibility to pneumonia and outcomes following infection are worse in males than females, both are worse with advancing age [31–33], and the development of pneumonia

increases the risk of repeat infections [34–36]. We assessed whether the increase in cytotoxic tau and A$\beta_{42}$ in CP patients reflected their sex and age (Fig 3). Increased cytotoxic tau and amyloids were more pronounced in males than females, although this analysis was not performed with reproductively-aged females compared to post-menopausal females, as this is beyond the scope of this study. Interestingly, cytotoxicity was increased in female CP patients compared to male and female CN patients. Tau was elevated in both older and younger CP patients, whereas A$\beta_{42}$ was elevated in younger CP patients. Whether female hormones (estrogens, progesterone) or androgens are involved in tau/amyloid release and cytotoxicity is unclear. Future studies to determine whether sex-linked hormones impact the relative cytotoxicity of tau and amyloids produced following infection are warranted. It is also possible that A$\beta_{42}$ and tau release are not always directly related to cytotoxicity and end-organ dysfunction. Other important injury mediators and their mechanism(s) are unclear at this time. Furthermore, we detected elevations in tau in all infected patients, whereas the increase in A$\beta_{42}$ was most evident in the younger population. As highlighted below, A$\beta_{42}$ was elevated in CP patients who survived compared to those that died. Thus, future studies should examine whether the increase in A$\beta_{42}$ during pneumonia fulfills a protective, antimicrobial role, as has been proposed [37]. Finally, tau, boiled tau, and A$\beta_{42}$ were increased in Caucasian CP patients compared to both Caucasian and minority CN patients. The difference between Caucasian CP and CN patients is consistent with our initial hypothesis that tau, boiled tau, and A$\beta_{42}$ lead to end-organ dysfunction. However, the basis for increases in tau, boiled tau, and A$\beta_{42}$ seen in Caucasian CP when compared to the minority CN groups is less clear. While there may be social, demographic, regional, or genetic differences between these groups that lead to differential susceptibility to bacterial pneumonia, this does not cover one important negative finding—there was no difference in tau, boiled tau, and A$\beta_{42}$ between minority CP and CN patients. The baseline levels of tau, boiled tau, and A$\beta_{42}$ in minority CN patients was similar to Caucasian CN patients; however, there was no significant increase in tau, boiled tau, and A$\beta_{42}$ when minority patients were infected with bacterial pneumonia. It is possible that minority patients do not elicit tau, boiled tau, or A$\beta_{42}$ after bacterial pneumonia. However, this possibility seems unlikely since large population studies indicate that minority patients may have delayed diagnosis of sepsis, of which bacterial pneumonia is a source [38]. It is possible that tau, boiled tau, and A$\beta_{42}$ increase and then decrease within a short time frame and, if care providers are delayed in diagnosing bacterial pneumonia, the increase seen in Caucasian CP patients might be missed. Future studies involving multiple clinical sites with a larger and more diverse patient population are warranted to address this important issue.

The development of end-organ dysfunction in critically ill patients increases morbidity and mortality [39–41]. Because pre-clinical studies indicate that tau and amyloids elicited secondary to bacterial infection cause injury to naïve cells and organs [13, 14, 16, 17], we analyzed end-organ dysfunction and mortality in our cohort. Increased levels of cytotoxic tau, boiled tau, and A$\beta_{42}$ in the bronchoalveolar lavage fluid were strongly associated with end-organ dysfunction (Fig 4, Table 3), but not mortality (Fig 5). This suggests that patients with increased tau and A$\beta_{42}$ may survive the initial insult, but chronic critical illness may follow with debilitating effects. Specifically, increased cytotoxic tau, boiled tau, and A$\beta_{42}$ corresponded with acute kidney injury, cardiovascular dysfunction, coagulopathy, and lung injury. This may indicate that, similar to pre-clinical studies [13, 14, 16, 17], bacterial pneumonia induces release of cytotoxic tau and amyloids that injure end-organs independently of the original infection. In this context, we provide evidence that bronchoalveolar lavage fluid concentrations of cytotoxic tau and A$\beta_{42}$ may serve as markers of organ dysfunction peripheral to the lung. While the focus of this exploratory study was the acute infection phase, survivors of critical illness have significant long-term sequelae, including cognitive dysfunction, arrhythmias, metabolic dysfunction, and

other organ injuries, collectively referred to as the post-ICU syndrome [42–44]. Future studies are imperative to examine the impact of cytotoxic tau and amyloids on long-term outcomes, especially since these cytotoxins can exhibit prion-like properties [13].

While cytotoxic tau and amyloids were overall increased in CP patients, compared to CN patients, the reason for the variability of the concentrations reported in this study is unknown. Pre-clinical studies indicate that bacterial virulence arsenals such as the Type III Secretion System of *P. aeruginosa* play a key role in communicating between bacteria and host cells to instigate the production of both cytotoxic tau and amyloid [13, 17]. However, the specific virulence mechanisms utilized by Gram-positive and Gram-negative bacteria to promote host cell cytotoxic tau and amyloid production are unknown. Genomic analyses to screen for the virulence arsenal expressed among bacterial species were not performed and will be a focus of future studies. Nonetheless, our study indicates that the generation of cytotoxic tau and $A\beta_{42}$ within the lung is equivalent across multiple bacterial species (*S. aureus*, *K. pneumoniae*, others; S1 Fig) and classifications (Gram-negative, Gram-positive; S2 Fig). While we have identified multiple bacterial species that are capable of eliciting cytotoxic tau and $A\beta_{42}$ production, future studies will examine whether other causes of pneumonia, including, but not limited to, viruses (i.e., influenza and SARS-CoV2), fungi (e.g., *Coccidioidomycosis*), and bacterial causes of community-acquired (e.g., *Streptococcus pneumoniae*) and atypical pneumonias (e.g., *Mycoplasma pneumoniae*) increase cytotoxic tau and $A\beta_{42}$.

The traditional paradigm holds that pneumonia is an acute cause of lung injury. However, it is increasingly recognized that clinical manifestations of pneumonia are neither acute nor localized to the lung—pneumonia can promote chronic disease that is evident in organs peripheral to the lung [45]. Furthermore, post-ICU syndrome causes long-term sequelae, including cognitive dysfunction, inability to rehabilitate, chronic infection, and poor quality of life [42–44]. How acute infection leads to chronic critical illness is an area of intense investigation. This exploratory study indicates that cytotoxic tau and $A\beta_{42}$ in the lung may be associated with distal organ injury, independent from the original pneumonia, and that cytotoxic tau and $A\beta_{42}$ in the bronchoalveolar lavage fluid portend poor outcomes. Together with the pre-clinical studies demonstrating the severe cytotoxicity induced by the infection-dependent tau and amyloids [13–17], our exploratory study elucidates new mechanistic possibilities into end-organ dysfunction during acute bacterial pneumonia and critical illness. Cytotoxic tau and amyloids may be considered novel biomarkers of bacterial pneumonia and the post-ICU syndrome.

In conclusion, we have performed a prospective, exploratory, controlled, observational clinical study demonstrating that cytotoxic tau and $A\beta_{42}$ levels are elevated in critically ill patients with bacterial pneumonia, and further, that the increase in these cytotoxins is associated with end-organ dysfunction. While the nature of this study is exploratory, it does implicate cytotoxic tau and amyloids in end-organ dysfunction and other pre-clinical studies (summarized in [15]) suggest multiple possible mechanisms. Clearly, this exploratory study is only the beginning of understanding how these new cytotoxins lead to acute and chronic end-organ dysfunction after acute bacterial pneumonia. Future investigations are needed to assess: 1) other organs or body fluids that may contain tau and amyloids after bacterial infection; 2) the longitudinal dissemination and excretion/decay of the tau and amyloids; 3) whether other infections cause release of tau and amyloids; and, 4) whether tau and amyloids can be used as a prognostic biomarker and/or be a target for prevention and/or treatment of chronic critical illness.

## Supporting information

**S1 Fig. Tau, boiled tau, $A\beta_{42}$, and cytotoxicity levels are similar in different bacterial species.** *(A)* Tau, *(B)* boiled tau, *(C)* $A\beta_{42}$, and *(D)* cytotoxicity were assayed as described in the

Methods in different species of bacteria: *Pseudomonas aeruginosa*, *Staphylococcus aureus*, *Klebsiella pneumoniae*, other single isolates, and other multi-microbial isolates. Data are expressed as mean ± SEM, *n* are listed below groups; * represents $p < 0.05$.
(TIF)

**S2 Fig. Tau, boiled tau, Aβ$_{42}$, and cytotoxicity levels are similar in different bacterial phenotypes.** *(A)* Tau, *(B)* boiled tau, *(C)* Aβ$_{42}$, and *(D)* cytotoxicity were assayed as described in the Methods in different phenotypes of bacteria: Gram-positive, Gram-negative, or both. Data are expressed as mean ± SEM, *n* are listed below groups.
(TIF)

**S1 File. Minimal data set.**
(XLSX)

## Author Contributions

**Conceptualization:** Phoibe Renema, Jean-Francois Pittet, Sarah Voth, Ron Balczon, Mike T. Lin, K. Adam Morrow, Jonathon P. Audia, Diego Alvarez, Troy Stevens, Brant M. Wagener.

**Data curation:** Phoibe Renema, Jean-Francois Pittet, Angela P. Brandon, Sixto M. Leal, Jr., Steven Gu, Grace Promer, Andrew Hackney, Phillip Braswell, Andrew Pickering, Grace Rafield, Sarah Voth, Jessica Bell, Troy Stevens, Brant M. Wagener.

**Formal analysis:** Phoibe Renema, Jean-Francois Pittet, Ron Balczon, Troy Stevens, Brant M. Wagener.

**Funding acquisition:** Sixto M. Leal, Jr., Sarah Voth, Ron Balczon, Mike T. Lin, Jonathon P. Audia, Diego Alvarez, Troy Stevens, Brant M. Wagener.

**Investigation:** Troy Stevens, Brant M. Wagener.

**Methodology:** Jean-Francois Pittet, Troy Stevens, Brant M. Wagener.

**Project administration:** Brant M. Wagener.

**Resources:** Brant M. Wagener.

**Supervision:** Brant M. Wagener.

**Validation:** Troy Stevens, Brant M. Wagener.

**Writing – original draft:** Phoibe Renema, Jean-Francois Pittet, Troy Stevens, Brant M. Wagener.

**Writing – review & editing:** Phoibe Renema, Jean-Francois Pittet, Angela P. Brandon, Sixto M. Leal, Jr., Steven Gu, Grace Promer, Andrew Hackney, Phillip Braswell, Andrew Pickering, Grace Rafield, Sarah Voth, Ron Balczon, Mike T. Lin, K. Adam Morrow, Jessica Bell, Jonathon P. Audia, Diego Alvarez, Troy Stevens, Brant M. Wagener.

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
