## [Decision Letter · Decision Letter 0]

6 Dec 2023

PONE-D-23-29603Tau and Aβ42 in lavage fluid of pneumonia patients are associated with End-Organ Dysfunction: A prospective exploratory studyPLOS ONE

Dear Dr. Wagener,

Thank you for submitting your manuscript to PLOS ONE. After careful consideration, we feel that it has merit but does not fully meet PLOS ONE’s publication criteria as it currently stands. Therefore, we invite you to submit a revised version of the manuscript that addresses the points raised during the review process.

We look forward to receiving your revised manuscript.

Kind regards,

Stephen D. Ginsberg, Ph.D.

Section Editor

PLOS ONE

Journal Requirements:

This study was supported by: 

B.M.W.--GM127584 and GM127584-S1; National Institute of General Medical Sciences; https://www.nigms.nih.gov

T. S. and R.B.--HL66299 and HL148069; National Heart, Lung, and Blood Institute; https://www.nhlbi.nih.gov

M.L., T.S., R.B.--HL140182; National Heart, Lung, and Blood Institute; https://www.nhlbi.nih.gov

J.P.A. and D.A.--HL118334; National Heart, Lung, and Blood Institute; https://www.nhlbi.nih.gov

S.M.L--AI170719; National Institute of Allergy and Infectious Diseases ; https://www.niaid.nih.gov

S.V.--HL147512, HL007778, REAP220049A0001; National Heart, Lung, and Blood Institute; https://www.nhlbi.nih.gov and Edward Via College of Osteopathic Medicine Research Eureka Accelerator Program (REAP); https://www.vcom.edu/research/research-strategic-plan

None of the sponsors or funders played any role in the study design, data collection and analysis, decision to publish, or preparation of the manuscript.

Additional Editor Comments:

Thank you for submitting your work to PLOS ONE. After careful consideration by 2 Reviewers and an Academic Editor, please make the corrections posed by Reviewers #1 and #2 so I can render a decision on this manuscript.

Reviewers' comments:

Reviewer's Responses to Questions

**Comments to the Author**

1. Is the manuscript technically sound, and do the data support the conclusions?

Reviewer #1: Yes

Reviewer #2: Yes

2. Has the statistical analysis been performed appropriately and rigorously? 

Reviewer #1: Yes

Reviewer #2: Yes

3. Have the authors made all data underlying the findings in their manuscript fully available?

Reviewer #1: Yes

Reviewer #2: No

4. Is the manuscript presented in an intelligible fashion and written in standard English?

Reviewer #1: Yes

Reviewer #2: Yes

5. Review Comments to the Author

Reviewer #1: Overall the scientific premise of the study is interesting, and the use of the patients from four intensive care units to address the hypothesis is appropriate. The references are appropriate and most recent. However, the study is required some refinement as follows.

Comment1: The abstract is well written, but I would like to see the number of patients included in Bacterial culture-positive and culture-negative groups as well as mean age and % of males in both groups in the abstract.

Comment2: In Introduction, provide the reference that describe the involvement of amyloid and tau in Parkinson's disease.

Also change Picks disease to frontotemporal tauopathies.

Comment3: There was difference in cytotoxic tau and amyloids levels by race which is interesting. I would also like to present this novel finding in the abstract as well as discuss in the discussion.

Comment 3: In table 1, add the SD for age in total patients and well as add the percent of male in total patients' column.

Comment 4: How data of smoking, Alcohol and Illicit Drug collected? were they self-reported?

Comment 5: Is the lung injury associated with amyloid and tau in CP group also common in male than females?

Comment 6: the cytotoxicity is more in female CP patients compared to both male and female CN patients even after male CP exhibited more amyloid and tau level than female CP. this is in contrast. I would suggest discussing in the discussion.

Reviewer #2: Introduction –

Two fundamental questions remain: how do A-beta and p-tau produced in the lungs reach the brain? The BBB is impermeable to macromolecular compounds. A-beta in the brain is deposited intravascular or in the parenchyma and NFT in the nerve cell. Moreover, compounds present in the cerebrospinal fluid reflect central pathology. It is also known that pathological changes in AD precede central pathology for many years and begin with the deposition of pathological proteins in the heart and other organs. I don't know if the authors made good research assumptions.

Hypotheses of the AD pathophysiology include amyloid cascade, inflammation, vascular, and infection factors. e.g. Grobler et al. PMID: 36442193,

Conclusion-

The use of A-beta and p-tau for treatment seems unattainable in the context of incurable neurodegenerative diseases.

Improve editorial errors.

6. PLOS authors have the option to publish the peer review history of their article (what does this mean?). If published, this will include your full peer review and any attached files.

Reviewer #1: **Yes: **Sonal Agrawal

Reviewer #2: No

---

## [Author Response · Author response to Decision Letter 0]

22 Jan 2024

All specific reviewer and editor comments have been responded to in the attached "Response to Reviewers".

---

## [Editor Report · Decision Letter 1]

31 Jan 2024

Tau and Aβ42 in lavage fluid of pneumonia patients are associated with end-organ dysfunction: A prospective exploratory study

PONE-D-23-29603R1

Dear Dr. Wagener,

We’re pleased to inform you that your manuscript has been judged scientifically suitable for publication and will be formally accepted for publication once it meets all outstanding technical requirements.

Kind regards,

Stephen D. Ginsberg, Ph.D.

Section Editor

PLOS ONE

---

## [Editor Report · Acceptance letter]

13 Feb 2024

PONE-D-23-29603R1 

PLOS ONE

Dear Dr. Wagener, 

I'm pleased to inform you that your manuscript has been deemed suitable for publication in PLOS ONE. Congratulations! Your manuscript is now being handed over to our production team.

Kind regards, 

on behalf of

Dr. Stephen D. Ginsberg 

Section Editor

PLOS ONE